# Hair Evaluation in Orthodontic Patients with Oligodontia

**DOI:** 10.3390/diagnostics14090945

**Published:** 2024-04-30

**Authors:** Małgorzata Zadurska, Adriana Rakowska, Ewa Czochrowska, Małgorzata Laskowska, Konrad Perkowski, Izabela Strużycka, Lidia Rudnicka, Agnieszka Jurek

**Affiliations:** 1Department of Orthodontics, Medical University of Warsaw, 02-097 Warsaw, Poland; malgorzata.zadurska@wum.edu.pl (M.Z.); ewa.czochrowska@wum.edu.pl (E.C.); malgorzata.laskowska@wum.edu.pl (M.L.); konrad.perkowski@wum.edu.pl (K.P.); 2Department of Dermatology, Medical University of Warsaw, 02-097 Warsaw, Polandlidia.rudnicka@wum.edu.pl (L.R.); 3Department of Comprehensive Dentistry, Medical University of Warsaw, 02-097 Warsaw, Poland; izabela.struzycka@wum.edu.pl

**Keywords:** hair abnormalities, isolated oligodontia, syndromic oligodontia, trichoscopy, trichogram

## Abstract

Oligodontia can be isolated or syndromic, associated with other ectodermal abnormalities. The aim of the study was to perform hair examination in orthodontic patients diagnosed with oligodontia with a low clinical expression of symptoms of ectodermal origin. All available orthodontic patients diagnosed with oligodontia in the permanent dentition were enrolled. Hair examination included clinical evaluation of the patients’ hair, trichoscopy, trichogram and evaluation of the hair shafts under a polarized light microscope. In total, 25 patients, 18 males and 7 females, aged 6 to 24 years were evaluated for the presence of dental and hair abnormalities. The number of congenitally absent teeth ranged from 6 to 24 teeth and diastemas, microdontia, taurodontism and altered tooth shape were found in 23 patients. Hair disorders were found in 68% of the subjects. Hypotrichosis, the heterogeneity of shaft color and loss of pigment, androgenetic alopecia, telogen effluvium, trichoschisis, pili canaliculi, trichorrhexis nodosa and pseudomoniletrix were observed. Trichoscopy and trichogram are valid non-invasive diagnostic tests which could be used to differentiate between isolated and syndromic oligodontia in patients with a low clinical expression of ectodermal symptoms.

## 1. Introduction

The term oligodontia (OMIM # 604625) describes the congenital absence of six or more teeth in the deciduous or permanent dentition or both, excluding third molars [1,2]. It is observed relatively rarely—in about 0.01–0.3% of the population [3,4]. Oligodontia is often accompanied by dental anomalies related to tooth morphology, position and time of eruption. Congenital absence of many teeth can lead to different types of malocclusion; often Class III malocclusion is seen, which may be attributed to the underdevelopment of maxillary alveolar process (Figure 1). The impaired smile aesthetics related to the presence of large spaces between teeth and tooth migration is a main reason for seeking dental rehabilitation including orthodontic treatment.

For example, regarding reduced tooth dimension and altered shape, incisors and canines most often have conical or peg-shaped clinical crowns and lateral incisors may have a reduced number of nodules on the incisor edge. Various dental anomalies, such as congenital absence and delayed eruption of deciduous and permanent teeth, tooth impaction, ectopic position of tooth buds, transposition of canines with lateral incisors and premolars, rotation of premolars, retained deciduous teeth and neonatal teeth have been described in patients with oligodontia [5,6,7]. Uslenghi et al. [8] demonstrated that in children with hypodontia, tooth development is, on average, delayed by 1.5 years compared to the control group with a full set of teeth. The delay is more significant in cases with multiple congenital tooth absence. The absence of tooth buds is the cause of underdevelopment of the maxillary dental arch and the dental portion of the mandibular arch, leading to the impaction of teeth in hypoplastic bone.

Oligodontia, similarly as hypodontia, can be isolated (non-syndromic) or syndromic and observed in various forms of ectodermal dysplasia with abnormal structures derived from the ectoderm and mesoderm. More than 190 distinct disorders have been described as ectodermal dysplasia in the literature [9].

Symptoms of ectodermal dysplasia involve the skin and its appendages (sweat, sebaceous, salivary, and tear glands, hair, nails), teeth, eye tissues, ears, adrenal glands, the nervous system and facial skull structures. In some syndromes, developmental disorders of ectodermal tissues are accompanied by abnormalities in other organs and systems, as well as intellectual underdevelopment. The pure form of ectodermal dysplasia is recognized when developmental disorders affect only the external germ layer.

If developmental disorders of ectodermal tissues are associated with abnormalities in other organs and systems, it is then diagnosed as an ectodermal syndrome. Ectodermal syndromes involve developmental disorders of mesodermal tissue, such as bone underdevelopment, polydactyly, syndactyly, ectrodactyly, defects in the urogenital system, underdevelopment of mammary glands and heart defects.

Over the past few decades, several classifications of ectodermal dysplasia have been developed to facilitate and refine diagnosis, prognosis and treatment planning. Examples include the classifications by Freire-Maia and Pinheiro, Priolo and Lagana and Lamartine [10,11,12]. In clinical practice, ectodermal dysplasias are divided into hypohidrotic and hyperhidrotic/anhidrotic types.

Hair disorders in patients with ectodermal dysplasia can be divided into several groups: congenital hypotrichosis, pigmentary disorders, congenital hair shaft dystrophy, acquired hair shaft dystrophy, hair cycle disorders and various types of alopecia.

The hair is often light, short, delicate, thin and brittle, sometimes with the absence of eyebrows and/or eyelashes. Described hair disorders include focal alopecia (e.g., in Bloch–Sulzberger syndrome), chronic inflammation of the hairy scalp with the formation of crusts (in Rapp–Hodgkin syndrome) and genetically determined hair shaft dystrophies (nodular hair cleavage, twisted hair, trichothiodystrophy). In patients with trichothiodystrophy, a metabolic sulfur defect results in very fragile and brittle hair. Biochemical studies show a reduction in the number of sulfur amino acids (cystine and cysteine).

In many forms of ectodermal dysplasia, certain symptoms allow for diagnosis shortly after birth. Sometimes it is only possible later in life if additional symptoms manifest or the disease appears in another family member. Genetic testing, although expensive, is possible in some forms of ED after determining the type of dysplasia and the gene suspected of mutation. The result of genetic testing does not impact the choice of treatment method, as it is symptomatic. It may influence the assessment of the risk of the defect occurring in siblings or offspring.

In clinical practice, the diagnosis of ectodermal dysplasia is made based on the presence of two or more abnormalities of structures originating from the ectoderm. Different forms of ED require differentiation from other ectodermal and mesoectodermal dysplasias. The prognosis depends on the form and clinical expression of ectodermal dysplasia. Generally, it is favorable, but the condition has an unfavorable impact on the patients’ quality of life.

The diagnosis of oligodontia can be performed by an orthodontist based on the presence of deciduous and permanent teeth in the oral cavity and teeth or tooth buds on radiographs. Congenital absence of primary teeth can be diagnosed in a three- to four-year-old child, while the absence of permanent teeth can be identified in a twelve- to fourteen-year-old child. The first sign is usually a delayed tooth eruption. Clinical examination reveals the absence of teeth in the oral cavity. Diagnosis is obtained through radiological examination, ruling out the presence of retained teeth. Based on radiological examination, it is possible to diagnose the absence of primary tooth buds shortly after birth and permanent teeth (excluding third molars) after the age of 6. Sometimes, the congenital absence of tooth buds is detected incidentally, based on X-rays taken for other reasons. Bergendal stated that oligodontia is most commonly diagnosed between the ages of 8 and 10 [4].

Nordgarden and co-authors evaluated 68 patients diagnosed with oligodontia and confirmed abnormalities in hair, nails and sweat glands in 57% of them [13]. In another Scandinavian study, one to four additional signs or symptoms from ectodermal organs besides oligodontia were found in 50% of their patients [4].

Congenital absence of tooth buds and the associated hypoplasia of dental arches affect the shape and dimensions of the palate and dental arches and contribute to the development of malocclusions. Malocclusions (most commonly skeletal Class III) and the congenital absence of tooth buds in patients with ectodermal dysplasia cause functional disturbances in the chewing organ. An abnormal resting tongue position, persistent infantile swallowing pattern, speech articulation disorders and difficulties in biting and chewing food are observed in these patients.

The aim of the study was to perform hair examination in orthodontic patients diagnosed with oligodontia who had a low clinical expression of the symptoms of ectodermal origin.

## 2. Materials and Methods

The study group included all available orthodontic patients diagnosed with oligodontia in the permanent dentition from files of the Department of Orthodontics, Medical University in Warsaw, Poland, for the last 20 years. Inclusion criteria included the following:-Age: >6 years old;-Congenital absence of at least 6 permanent teeth or tooth buds;-No clinical symptoms of onychodysplasia,-No clinical symptoms of dyshidrosis;-No other congenital dentofacial deformities, e.g., cleft lip or palate;-Absence of other anomalies accompanying ectodermal syndromes, such as polydactyly, syndactyly or ectrodactyly.

Anamnesis, extraoral and intraoral clinical examination, panoramic and cephalometric radiographs and diagnostic orthodontic models were collected in all patients. The dental status and the number of congenitally missing teeth was assessed during intraoral clinical examination and radiological assessment of panoramic radiographs.

Hair examination was performed by two of the authors (A.R., L.R.). It comprised three types of non-invasive hair examinations, which were applied in previous reports [14,15,16,17]:Trichoscopy, allowing examination of the hairy skin using a dermoscope with computer software and a digital camera; the optical system allows viewing of lesions at magnifications of 20–70 times and archiving of images, and the examination does not require hair plucking.Trichogram, which is the most widely used method in trichology, allows assessment of whether the proportions of hair in appropriate growth phases are maintained; the examination involves microscopic evaluation of about 100 hair shafts from the patient’s head taken from two or four areas and then examining the prepared hair specimens under a microscope. The microscopic examination assesses the hair growth phase based on its shape, shaft color, presence of the hair shaft sheath, and angle of hair bending in relation to the shaft. It enables evaluation of the number of hairs in different phases: anagen, catagen, and telogen, as well as the number of dystrophic/dysplastic hairs.Examination under polarized light, allowing assessment of the cortex and shaft structure of the plucked hair, which is used in cases of suspected congenital hair shaft disorders (trichothiodystrophy, monilethrix, and pili annulati) and increased hair fragility [18,19].

Two weeks before the trichological examinations, patients could not undergo any hair procedures (e.g., dyeing or perming), while four days before they were not allowed to wash their hair.

During trichoscopic examination, which is based on the videodermoscopic evaluation of hair and scalp, hair shafts were evaluated. Dermoscopy is a method that allows the observation and assessment of structures at the skin surface, the skin-epidermal junction, the upper layers of the dermis, and hair using a surface microscope called a dermoscope. Dermoscopes are optical devices equipped with side lighting for the examined surface and provide magnification ranging from 20 to 70 times. In addition to magnification, dermoscopy visualizes structures that are not visible to the naked eye. The main advantages of the method include non-invasiveness, the ability to visualize hairs in their natural environment, and the observation of ongoing pathophysiological processes in real-time [20,21].

In particular, their color, thickness, presence of congenital hair shafts dystrophies and percentage of follicular units with one, two or three hairs were assessed. Beside hair shafts trichoscopy enables evaluation of follicular openings, skin blood vessels and background color or scaling.

The trichogram was used to assess the hair cycle (anagen, catagen and telogen). It involves examination of 100 hair bulbs of a patient under a microscope, which are plucked from various areas of the hairy scalp. The specific percentage distribution of different phases of the hair cycle is representative for the entire scalp, and identified deviations from the norm can be helpful in directing further investigation of hair loss.

In a healthy patient, about 85% of hair should be in the growth phase (the anagen phase), less than 1% in the transition phase (catagen phase) and about 15% in the resting phase (telogen phase). A trichogram in which at least 80% of hair is in the anagen phase is considered normal. Polarized light testing with 90-degree polarizing filters crossed at 90 degrees was used to reveal abnormalities in the shaft structure.

Ethical approval was obtained from the Ethical Committee of the Medical University in Warsaw, Poland (KB-014/09). All the patients have signed an informed consent for the participation in the study (in minor patients, the consent was signed both by a parent and a patient).

## 3. Results

The study group consisted of 25 patients with oligodontia in the permanent dentition, 18 males and 7 females, aged 6 to 24 years. A pediatric or a family doctor did not diagnose the presence of ectodermal dysplasia in any of the included patients. Also, none of the patients had a confirmed diagnosis of the ectodermal dysplasia or had a genetic test for its presence. All patients were healthy in general.

The number of congenitally absent teeth, which were assessed based on a clinical and radiological examination ranged from 6 to 24 teeth (Table 1). Dental anomalies were found in 23 patients, 16 males and 7 females. The most common dental anomalies were associated with reduced size of teeth such as microdontia, narrow and conical teeth and abnormal tooth shape. The reduced number of teeth resulted in the presence of diastemas and spaces between teeth in many of the examined patients. In particular microdontia was observed in 9 patients (7 males and 2 females), diastemas in 5 patients (3 males and 1 female), taurodontism in 5 patients (4 males and 1 female), ectopic canine position in 4 patients (1 male and 3 females), conical shape of canine in 2 patients (2/25, 1 male and 1 female) transposition of canine with first premolar in 1 patient (1/25, 1 male), abnormal shape of maxillary central incisors in 3 patients (3/25, 2 males and 1 female), narrow clinical crowns of the maxillary incisors in 3 patients (3/25, 3 males), elongated clinical crowns of the mandibular first premolars in 1 patients (1/25, 1 male), and tricuspid lower second molars in 1 patients (1/25, 1 male).

Different types of hair disorders were found in 17 (68%) of the patients—13 males and 4 females (Table 2). Hypotrichosis was diagnosed in 11 patients (44%) and mainly the diagnosis included a predominance of pilosebaceous units with single hair in trichoscopy. In one patient (4%), androgenetic alopecia was diagnosed in the frontal area. Hair shafts thickness heterogeneity, loss of follicular units with three hairs, yellow dots and telogen effluvium were observed in five patients (20%). Normally, follicular units with three hairs comprise more than 50% of the follicular units. The abnormalities of hair shaft structure included trichoschisis in two patients (8%), pili canaliculi in one patient (4%), trichorrhexis nodosa in one patient (4%) and pseudomoniletrix in one patient (4%) (Figure 2 and Figure 3). Hair shaft pigmentation disturbances were found in 52% of patients such as the heterogeneity of shaft color in 3 patients (12%), loss of hair shaft pigmentation in 10 patients (40%) and cases in which more than 10% of hair shafts were revealed. There were no cases with pili torti and trichothiodystrophy observed in patients with a high clinical expression of ectodermal dysplasia. In patients with trichothiodystrophy due to a metabolic sulfur defect, hair is found to be very brittle and prone to breaking. Biochemical studies reveal a decrease in the number of sulfur-containing amino acids (cystine and cysteine) [22].

## 4. Discussion

Oligodontia is a phenomenon observed much less frequently than hypodontia. The low occurrence rate of oligodontia, estimated to be between 0.01% and 0.3% of the population, results in studies being conducted on relatively small samples [1,3,4,13,23,24]. In the present study, 25 patients with oligodontia were selected from a sample of 180 oligodontia patients, in whom ectodermal dysplasia was suspected but not previously diagnosed by pediatricians and family doctors. Hair examination was only performed in the study sample because not all patients were available for the examination, as the original sample was collected over 20 years. Thus far, no studies have been published in the literature on hair evaluation in oligodontia patients. The results of the study, which confirmed the higher occurrence of hair abnormalities in patients with oligodontia, should encourage future investigations in larger samples with isolated and non-isolated oligodontia.

Methods for examining hair include clinical examination, pull test, trichoscopy (videodermoscopy), trichogram, examination under polarized light microscopy, examination under an electron microscope and vertical and horizontal skin biopsy with histopathological examination, as well as immunopathological tissue examination. In the present study, in addition to clinical examination, non-invasive hair examinations were used (trichoscopy, trichogram and examination under polarized light). It was confirmed that two-thirds of the patients with oligodontia in the permanent dentition included in the study sample had abnormal hair morphology and color. Disturbances in the structure of the hair stems are rare in the general population, and their occurrence is always genetically determined, usually as an ectodermal dysplasia or syndrome. People under 18 years of age do not have any abnormalities in the color of the hair shaft, such as gray hair. The shift in the hair cycle may have other causes, but then it is not a chronic disorder but a temporary one.

The most commonly observed symptoms were hair shaft pigmentation abnormalities, hypotrichosis and telogen effluvium (short anagen syndrome). Structural abnormalities of the hair shaft were less common. The diagnosis of androgenetic alopecia in one patient has been recognized as a comorbid disease. None of the assessed patients were diagnosed with the most severe hair disorder—trichothiodystrophy. It is usually present in patients with ectodermal dysplasia, mainly in the hypohidrotic form. The abnormalities of the hair shaft associated with a defect in sulfur metabolism make the hair brittle and breakable. Biochemical studies show a reduced number of sulfur amino-acids, cystine and cysteine. Half of the patients with trichothiodystrophy suffer from photosensitivity [25,26].

The clinical examination, trichoscopy, trichogram and polarized light examination in a group of sixteen patients with pronounced clinical symptoms of ectodermal dysplasia conducted by Rakowska et al. demonstrated the presence of all previously described hair abnormalities in the literature [27]. The study conducted by Zadurska did not show a correlation between the number of missing permanent tooth buds and hair abnormalities; however, clinical observations indicate that in hypohidrotic ectodermal dysplasia, there is a more extensive absence of permanent tooth buds than in cases of isolated oligodontia [23].

The diagnosis of syndromic oligodontia is not difficult in cases with distinct symptoms from hair, onychodysplasia or sweating disorders. A diagnostic problem to differentiate between systemic and non-systemic oligodontia may occur when clinical signs are discrete. According to Dhamo et al., in patients with syndromic oligodontia the following dental anomalies are observed more often than in non-syndromic oligodontia: agenesis of the central incisors and second molars, mandibular lateral incisors, alterations in the shape of incisors and canines and a delayed development of permanent teeth [14]. The presence of such tooth abnormalities was confirmed in our patients (Table 1).

The diagnosis of ectodermal dysplasia is generally confirmed by genetic testing, if the gene for the particular form of dysplasia has been detected. Numerous genes have been identified whose mutations cause abnormal ectoderm–mesenchyme interactions at the molecular level and are responsible for the ectodermal dysplasia phenotype. These genes include *PAX9*, *EDA*, *EDAR*, *XEDAR*, *NEMO*, *EDARADD*, *WNT10A*, *AXIN2* and *LEF1* [28,29,30,31,32]. Song et al. found, that the EDA gene is responsible not only for ectodermal dysplasia, but also for non-syndromic oligodontia in boys [33]. Also, Reinhold et al. reported that the EDA gene in female patients with ectodermal dysplasia is responsible for severe tooth malformations [34]. Findings regarding the frequency of oligodontia by gender are inconsistent. In the study sample, two-thirds of the examined participants were boys.

However, genetic factors are responsible for only approximately 50% of ED disorders [10]. Genetic testing confirming the presence of ectodermal dysplasia was not performed in any of our patients because of the low clinical expression of the abnormalities in the structures of ectodermal origin. The classification of ectodermal dysplasia according to Freire-Maia and Pinheiro is based on the assessment of the presence of four basic clinical symptoms, marked with numbers: 1—hair abnormalities, 2—dental abnormalities, 3—nail dysplasia (onychodysplasia), 4—sweating disorders (dyshidrosis) [10]. Two of the abnormalities mentioned above are sufficient to diagnose an ectodermal defect [35].

In patients with ectodermal dysplasia, numerous hair disorders have been described: hypotrichosis (in a trichoscopic examination seen as increased percentage of follicular units with one shaft), telogen effluvium, congenital hair shafts abnormalities such as pili torti, trichoschisis, pili canaliculi and trichothiodystrophy. In addition, heterogeneity of shaft thickness, trichorrhexis nodosa and pseudomoniletrix were noted in single cases. One of the most common findings regarding hair shafts in patients with ectodermal dysplasia was hair shafts pigmentation abnormalities. In the child population, the presence of over 10% of gray hair and heterogeneity of the color of shafts have been described [27,36].

The results of the present study confirmed the presence of hair abnormalities in many of the patients with oligodontia in the permanent dentition which should indicate the necessity for further diagnostic evaluation towards the presence of ectodermal dysplasia. Genetic testing would be recommended to confirm the presence of syndromic and non-syndromic oligodontia in our sample. If positive, trichoscopy and trichogram which are non-invasive hair examinations, could be advised in all oligodontia patients to select those who should be further diagnosed for the presence of ectodermal dysplasia. An orthodontist or a pediatric dentist, who routinely diagnoses the presence of teeth or tooth buds during clinical and radiological dental examination, could play a pivotal role in diagnosis of syndromic and non-syndromic oligodontia.

The limitations of the present study include the small sample size and rare and non-homogenous characteristics of hair abnormalities symptoms. Thus, it seems very important to search for clear clinical criteria for the diagnosis of hair abnormalities associated with the presence of oligodontia. The low prevalence of oligodontia in the population makes it difficult to perform a study comprising a large number of patients. Multicenter studies could help to further evaluate possible correlations between dental and hair abnormalities. Comparative hair analysis with a healthy control group should be considered to confirm the higher incidence of hair alterations in patients with oligodontia.

## 5. Conclusions

Hair abnormalities were often found in patients with oligodontia in the permanent dentition included in the present study. The most common were the presence of hypotrichosis and gray hair and the shifting of the hair cycle towards telogen. Trichoscopy and trichogram are valid non-invasive diagnostic tests which could be used to differentiate between isolated and syndromic oligodontia in patients with a low clinical expression of ectodermal symptoms.

## Figures and Tables

**Figure 1 diagnostics-14-00945-f001:**
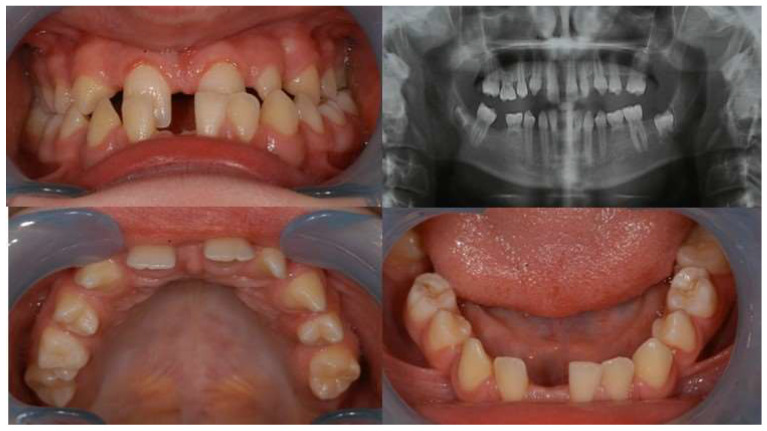
Intraoral photographs and a panoramic radiograph of the patient with oligodontia in the permanent dentition and Class III malocclusion.

**Figure 2 diagnostics-14-00945-f002:**
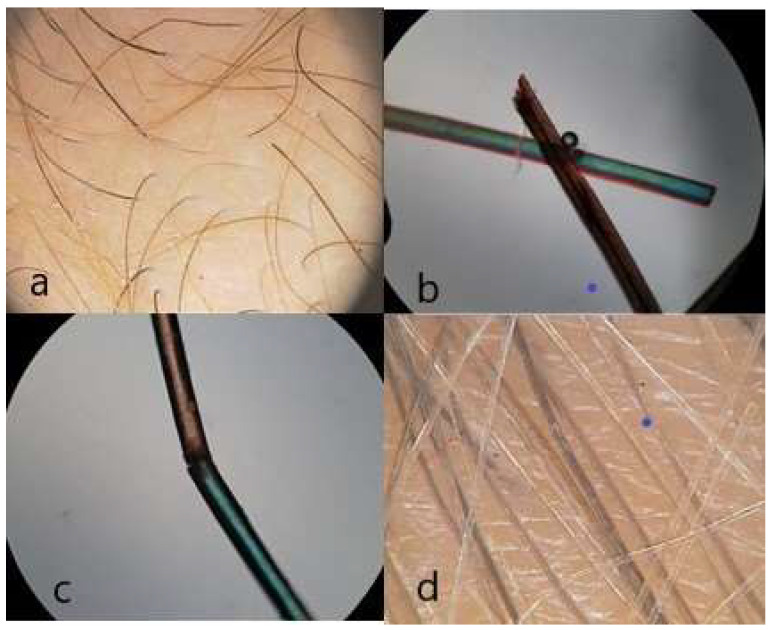
Hair abnormalities: (**a**) hypotrichosis seen as predominance of follicular units with single hair (trichoscopy, ×20), (**b**) pili canaliculi (microscopy; ×40), (**c**) *trichoschisis*, (microscopy; ×40) (**d**) heterogeneity of hair shaft pigmentation (trichoscopy; ×50).

**Figure 3 diagnostics-14-00945-f003:**
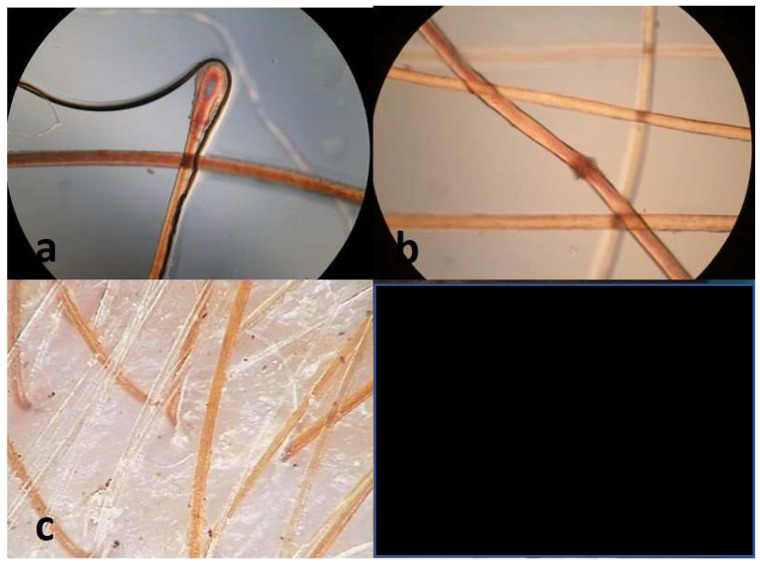
Hair abnormalities: (**a**) telogen bulb in microscopic examination with polarized. Light (microscopy; ×40), (**b**) nodular trichorrhexis (microscopy; ×40), (**c**) pigment loss (over 10% of gray hairs) (trichoscopy ×50).

**Table 1 diagnostics-14-00945-t001:** Dental and hair characteristics of 25 patients with oligodontia in the permanent dentition.

No	Gender	Number of Missing Teeth	Dental Anomalies	Hair Abnormalities
1	M	22	Microdontia of the maxillary right central incisor	Hypotrichosis
2	M	21	Diastema	Not present
3	M	6	Diastema	Shifting the hair cycle towards telogen
4	M	6	Diastema, ectopic position of the mandibular canines	Hypotrichosis, more than 10% of hair without dye. Trichoschisis, shifting the hair cycle towards telogen
5	F	9	Abnormal shape of the maxillary central incisors and conical shape of the canines	Not present
6	F	7	Ectopic position of the maxillary right canine	Hypotrichosis, shaft color heterogenity
7	M	11	Not present	More than 10% of hair without dye. Shaft color heterogenity
8	F	10	Microdontia of the maxillary left lateral incisor	More than 10% of hair without dye
9	M	7	Abnormal shape of the maxillary central incisors, microdontia of the maxillary lateral incisors and taurodontism of the first molars	More than 10% of hair without dye
10	M	11	Narrow clinical crowns of the maxillary incisors, taurodontism of the first molars and maxillary second molars	Not present
11	M	15	Microdontia of the maxillary left lateral incisor and upper canines	Hypotrichosis, more than 10% of hair without dye
12	M	13	Microdontia of the maxillary left lateral incisor and transposition of the right canine and first premolar in the mandible	Hypotrichosis, more than 10% of hair without dye. Shifting the hair cycle towards telogen
13	M	17	Abnormal shape of the maxillary central incisors	Hypotrichosis, more than 10% of hair without dye. Trichorrhexis nodosa
14	M	14	Microdontia of the maxillary incisors	Not present
15	M	26	Conical shape of the maxillary lateral incisors	Hypotrichosis, shifting the hair cycle towards telogen. Trichoschisis
16	M	13	Microdontia of the maxillary left incisors and all premolars	Not present
17	M	12	Narrow crowns of the maxillary central incisors, elongated crowns of the mandibular first premolars, taurodontism of the first molars and mandibular second molars	Not present
18	F	8	Taurodontism of the mandibular first molars and ectopic position of the maxillary canines	Not present
19	F	15	Diastema, ectopic position of the mandibular canines	Hypotrichosis, shaft colour heterogenity
20	M	24	Tricuspid lower second molars	Hypotrichosis, androgenic alopecia
21	F	9	Diastema	Not present
22	M	11	Not present	Pseudomoniletrix, pili canaliculi
23	M	10	Narrow clinical crowns of the maxillary incisors, taurodontism of the first molars	Hypotrichosis, more than 10% of hair without dye. Shifting the hair cycle towards telogen
24	F	9	Microdontia of the maxillary lateral incisors, transposition of the mandibular canines and first premolars	Hypotrichosis, more than 10% of hair without dye
25	M	6	Microdontia of the maxillary lateral incisors	More than 10% of hair without dye

**Table 2 diagnostics-14-00945-t002:** Hair abnormalities in 25 patients with oligodontia in the permanent dentition.

No	Hair Abnormalities	Number and Percentage of Patients (25 in Total)
1	Hypotrichosis	11 (44%)
2	Trichoschisis	2 (8%)
3	Pili canaliculi	1 (4%)
4	Trichorrhexis nodosa	1 (4%)
5	Pseudomoniletrix	1 (4%)
6	Shifting the hair cycle towards telogen	5 (20%)
7	Androgenic alopecia	1 (4%)
8	Shaft color heterogeneity	3 (12%)
9	More than 10% of hair without dye	10 (20%)

## Data Availability

Raw data are available from the corresponding author on reasonable request.

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
