# Peer review of "Hair Evaluation in Orthodontic Patients with Oligodontia"

_diagnostics, 2024, doi:10.3390/diagnostics14090945_

Round 1
Reviewer 1 Report
Comments and Suggestions for Authors
- Please provide information on who performed the trichoscopy and trichogram examination
- More basic literature on the technique and interpretation of trichoscopy and trichogram should be provided.
- More photographs of dental and capillary alterations would be advisable.
- As the sample is small (25 patients, 18 males and 7 females) a healthy control group should be included to compare the results, as some pathologies are very prevalent at that age (e.g. telogen effluvium).
- For the same reason, it should be convenient to include another sample of patients with oligodontia who had a high clinical expression of the symptoms of ectodermal origin.
Comments on the Quality of English LanguageGrammar and some sentences should be improved.
Author Response
Dear Reviewer,
We are very grateful for your time and valuable comments on our paper.
In particular regarding your comments:
- Please provide information on who performed the trichoscopy and trichogram examination
We have included in the paper that 2 of the authors have performed the examination
- More basic literature on the technique and interpretation of trichoscopy and trichogram should be provided.
We have added the references.
- More photographs of dental and capillary alterations would be advisable.
We have added clinical photographs of the dentition of one of the patients.
- As the sample is small (25 patients, 18 males and 7 females) a healthy control group should be included to compare the results, as some pathologies are very prevalent at that age (e.g. telogen effluvium).
This is a very good suggestion for the future studies and we have included this information in limitations.
- For the same reason, it should be convenient to include another sample of patients with oligodontia who had a high clinical expression of the symptoms of ectodermal origin.
We have included in the discussion the prevalence of oligodontia in the population and problems regarding the small sample sizes in studies on oligodontia. We have also stated that: in the present study, twenty-five patients with oligodontia were selected from a sample of 180 oligodontia patients, in whom ectodermal dysplasia was suspected but not previously diagnosed by pediatricians and family doctors. Hair examination was only performed in the study sample because not all patients were available for the examination as the original sample was collected over 20 years. We have also added in the limitations that the sample size was small.
Reviewer 2 Report
Comments and Suggestions for Authors
This is an interesting prospective study in a rarely assessed population. Repeating this assessment in a different population has value.
Table 1 could benefit from adding the hair abnormalities found for each of the subjects. If they can be grouped together either according to dental or hair findings or both, this would aid readability.
This reviewer realizes the sample size is small. Were specific dental findings more predictive of finding hair abnormalities? Which ones were associated and which ones were not associated?
Author Response
Dear Reviewer,
We are very grateful for your time and valuable comments on our paper.
In particular regarding your comments:
Table 1 could benefit from adding the hair abnormalities found for each of the subjects. If they can be grouped together either according to dental or hair findings or both, this would aid readability.
We have added the hair abnormalities to the Table 1.
This reviewer realizes the sample size is small. Were specific dental findings more predictive of finding hair abnormalities? Which ones were associated and which ones were not associated?
The dental characteristics of all patients is described in details in the Table 1. We have added that: the most common dental anomalies were associated with reduced size of teeth such as microdontia, narrow and conical teeth and abnormal tooth shape. The reduced number of teeth resulted in the presence of diastemas and spaces between teeth in many of the examined patients.
Round 2
Reviewer 1 Report
Comments and Suggestions for Authors
The content of the article has been significantly improved by the authors' contributions, mainly material and methods, description of hair abnormalities, photographs and references. At present, the small sample size and the absence of a healthy control group cannot be solved to obtain more relevant conclusions, therefore we suggest the authors to work on these shortcomings for future publications.